



# Forest conversion reduces soil water retention in tropical rainforest by altering soil properties

Qiaoyan Chen[1], Siyuan Cheng[1], Shuting Yu[1], Xiaowei Guo[2], Zhongyi Sun[1], Zhongmin Hu[1], Licong Dai[1]*

[1]Key Laboratory of Agro-Forestry Environmental Processes and Ecological Regulation of Hainan Province, Hainan University, Haikou, 570228, China
[2]Qinghai Provincial Key Laboratory of Cold Regions Restoration Ecology, Northwest Institute of Plateau Biology, Chinese Academy of Sciences, Xining 810001, China

*Correspondence to*: Licong Dai (licongdai@hainanu.edu.cn)

**Abstract.** Extensive primary forests are being converted to secondary forests and plantation owing to human activities in recent decades, which has substantial effects on soil hydrological processes. However, the potential impact of forest conversion on soil water retention remains poorly understood. In this study, tropical primary forests (PF), secondary forests (SF) and rubber monocultures (RM) converted from tropical primary forests were selected on Hainan Island, to examine the variation in soil water retention across three forest types and their controlling factors. We found that the primary forests exhibited significantly greater water retention capacity than secondary forests and rubber monocultures. However, secondary forests showed higher water retention than rubber monocultures in shallow soils but lower in deep soils. Similarly, primary forests demonstrated significantly greater soil water storage capacity than secondary forests and rubber monocultures, but secondary forests and rubber monocultures had obvious seasonal variations, which showed that secondary forests had a higher water storage capacity than rubber monocultures in the rainy season, and display opposite pattern in the dry season. The saturated hydraulic conductivity in primary forests was higher than that in secondary forests and rubber monoculture. Furthermore, forest types influenced soil properties, with secondary forests and rubber monoculture showing higher bulk density but lower soil capillary porosity compared with primary forests. Among all factors, soil porosity emerged as the dominant controller of water retention, where total porosity and capillary porosity accounted for 31.49% and 30.61% of variation respectively, while soil bulk density contributed relatively less (12.46%). These findings indicate that the conversion of tropical primary forests to secondary forests and rubber monocultures is detrimental to soil water retention and storage. Our results can provide scientific insights for forest development and management in the tropical rainforest.

## 1 Introduction

Rapid economic development and population growth in tropical regions have accelerated the depletion of natural resources, and the transition from primary tropical forests to secondary forests and plantations has occurred or is occurring throughout the tropical regions (Li et al., 2012; Sidle et al., 2006). The area of natural tropical forests decreased by 11%, whereas the area of plantations increased by 87% during 1990-2015 (Keenan et al., 2015). Forest restoration strategies following the destruction of primary tropical forests include non-native plantation forests established on the original land and naturally



restored secondary forests (Jones et al., 2022). However, degradation to either secondary forests or plantations results in reduced biodiversity and altered soil nutrient cycling (Peng et al., 2022). Moreover, forest conversion due to changing vegetation conditions has important effects on soil properties, which in turn affect soil water retention (Mahe et al., 2005; Templer et al., 2005; Hoek van Dijke et al., 2022). Therefore, exploring the effects of forest conversion on soil water retention is an important guide for tropical rainforest management.

Soil water retention is a critical parameter for the hydrological process, controlling soil water availability, storage dynamics and soil-atmosphere exchange process (Geroy et al., 2011; Ebel, 2012) , which is greatly influenced by forest conversion via altering vegetation structure and soil characteristics (Wang et al., 2013). Due to higher species richness and complex root systems, soils in primary forests tend to be looser and accumulate more organic matter, thereby having higher soil water retention capacity (Lüscher & Zürcher, 2003). However, in recent decades, economic development and slash-and-burn cultivation by ethnic minorities have led to extensive degradation of primary forests, naturally restored secondary forests and commercial monocultures (rubber and betel monocultures) have become the main forest types on Hainan Island (Guangyang et al., 2023). The forest conversion is expected have substantial impacts on soil properties and soil hydrological aspects. On the one hand, it causes changes in forest structure, which affects canopy interception, leading to changes in soil hydrological processes (Sutanto et al., 2012). On the other hand, shifts in forest type affect soil properties, which in turn affect soil water retention (Haghighi et al., 2010). Soil water retention is affected by multiple factors, including soil physicochemical properties (e.g., organic matter content, soil texture, and soil structure) (Launiainen et al., 2022; Zhou et al., 2020; Otalvaro et al., 2016; Yang et al., 2014), dry and wet alternation cycles (Kercheva et al., 2019), and plant root systems (Gao et al., 2018) among others. Previous studies have pointed out that rubber monocultures exhibit limited biodiversity, reduced carbon sequestration capacity and severe soil erosion (Ziegler et al., 2009). The conversion of primary forest to secondary forest and rubber monoculture leads to a decline in soil physicochemical properties and nutrients, affecting soil water retention capacity (Ahrends et al., 2015; Deuchars et al., 1999). $\alpha$- and $\beta$-diversity in rubber monocultures are lower than in tropical rainforests due to monoculture canopy and intensive management (Kerfahi et al., 2016). The complexity of plant, plant litter and root interactions in natural rainforests results in rubber monoculture having significantly lower soil porosity and significantly higher soil bulk density than tropical primary and secondary forests (Sun et al., 2021). By evaluating the hydrological properties of old-growth forests, early stages of primary forest succession, and managed plantations, Jones et al. (2022) found that old-growth forests have higher evapotranspiration and stable water yield, whereas managed plantations have lower water yield, and that early stages of primary forest succession may provide higher water yield compared to managed plantations. In addition, van der Sande et al. (2022) found a sharp increase in soil bulk density and a decrease in soil total carbon and nitrogen after deforestation, which may be related to the reduction in litter inputs and a sharp decrease in fine root biomass resulting from the deforestation of primary forests. Overall, considering a large number of primary forests have been transformed into secondary and plantation forests, the effects on their soil water retention are not yet clear, and studies on the seasonal variation of soil water storage are lacking.



To fill these knowledge gaps, this study selected tropical primary forests, naturally restored tropical secondary forests, and rubber monoculture forests in Hainan Province, and measured soil and hydrological properties, aiming to investigate the changes in soil hydrological processes under different forest types. Specifically, we aim to (1) examine the effects of the conversion of primary forest to naturally restored secondary forest and rubber monoculture on soil water retention, and (2) reveal the main factors affecting soil water retention in the three forest types. By comparing the soil properties and hydrological functions of the three forest types, this study explores the effects of different forest types on soil ecological functions in the tropics, with a view to providing a scientific basis for sustainable forest management.

## 2 Materials and methods

### 2.1 Study site

We performed the study in Baoting County, located on Hainan Province, China (Fig. 1), a region characterized by a tropical monsoon climate with marked seasonal climatic patterns. The area experiences distinct dry (November to April) and rainy (May to October) seasons, receiving approximately 1900 mm average annual precipitation with more than 80% occurring during the rainy season (Sun et al., 2017). The average annual temperature remains relatively constant at 23 °C (Chen et al., 2024). The predominant soils in the study area are ferralsols and primitive soils, classified according to the USDA Soil Taxonomy (Sun et al., 2021). Over the past decades, slash-and-burn practices and economic development have led to extensive destruction of primary forests. Currently, naturally regenerated tropical secondary forests and plantations (rubber and betel nut plantations) constitute the major forest types in Baoting. In addition, some patches of primary forest have been preserved, providing a natural experimental setting for investigating the soil water retention capacity and its controlling factors across different tropical forest types. Additional details on the three forest types are provided in Table 1.

Table 1 Basic information of three forest types

| Forest types | Height (m) | DHB (cm) | Age (year) | Vegetation coverage (%) | Litter yield (t·hm$^{-2}$) |
|---|---|---|---|---|---|
| Primary forests | 17.79±1.47 | 28.82±3.53 | >80 | 91 | 4.27±1.95 |
| Secondary forests | 11.16±0.43 | 22.96±1.95 | 20~30 | 80 | 2.08±0.15 |
| Rubber monoculture | 12.38±0.21 | 20.56±1.40 | 20~30 | 50 | 1.66±0.11 |

Note: DHB: diameter at breast height; data expressed as mean ± se.



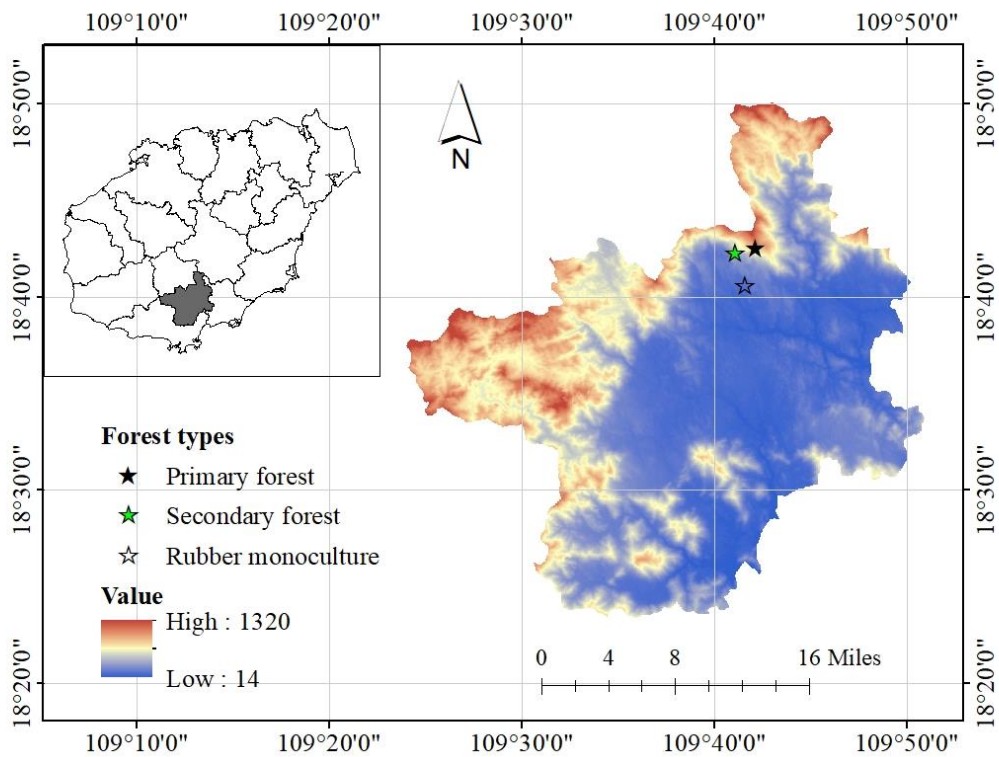

Figure 1 Study area and sample plots of three forest types (i.e. primary forests, secondary forests, and rubber monocultures).

## 2.2 Study Method Design and Sample Acquisition

The field investigations were carried out from January 2023 to December 2024 in Baoting County, Hainan Province, China. To assess alterations in soil hydrological dynamics resulting from tropical forest conversion, we implemented a chronosequence approach using space-for-time substitution. The study focused on three characteristic forest types in the region: (1) tropical primary forests (PF); (2) naturally regenerated secondary forests (SF); and (3) commercial rubber plantations (RM) (Fig. 1). Notably, to ensure comparability among the study sites, all selected plots had similar biophysical conditions, including altitude, slope, and aspect. In the rubber monoculture sites, soil sampling was conducted in the interrow areas to minimize the influence of management activities on soil structure. As each site with a specific forest type has uniform soil parent material, topographic, and climate conditions, our assessment target can rule out the impacts of these environmental variables and solely mirror the impacts of the forest type. Consequently, our findings will be capable of more precisely evaluating the differences in the effects of various forest types distributed across different sites on soil hydrological processes.

For each forest type, three study sites were established. Within each site, three plots (20 m × 20 m) were established, with four randomly positioned subplots (1 m × 1 m) per plot. Soil sampling included both disturbed and undisturbed samples from



each subplot. Undisturbed soil cores (100 cm³) were extracted in situ at six depths (0-10, 10-20, 20-30, 30-40, 40-50, and 50-60 cm) using a ring knife sampler for measurements of soil water retention characteristics, soil porosity, soil bulk density, and soil water storage. Additionally, disturbed samples were collected with a soil auger for laboratory analysis of soil physiochemical properties, such as total nitrogen, total carbon, soil organic matter content, soil particle size distribution, soil available phosphorus, and soil available nitrogen.

**2.3 Study Method Design and Sample Acquisition**

Soil particle composition was measured with reference to (Peng et al., 2014), and classified as clay (0-0.002 mm), silt (0.002-0.02 mm), and sand (0.02-2 mm), according to international classification standards. Soil organic matter (SOM) was measured following (Nelson & Sommers, 1982). For elemental composition, we used an elemental analyzer to determine total nitrogen content (TN) and total carbon content (TC), subsequently calculating C:N ratios. Nutrient availability assessments included available nitrogen (AN) measurement by alkali solution diffusion method and available phosphorus (AP) determination through molybdenum-antimony anti-spectrophotometric method (Soil Science Society of China, 2000).

The soil hydrology experiment was performed using undisturbed soil core samples following a standardized protocol. Initial sample weights were recorded before conducting sequential water retention measurements. Samples were first saturated for 24 hours until constant weight was achieved, followed by drainage measurements after 2-hour and 24-hour periods in dry sand. Finally, samples were oven-dried at 105 ℃ for 24 hours until constant weight was achieved to determine the dry soil mass (Wds, g). Following this approach, saturated water holding capacity (SWHC, %), capillary water capacity (CWC, %), field capacity (FC, %), soil water content (SWC, %) were determined (Chen et al., 2024). The calculated parameters included: bulk density and soil porosity (total, non-capillary, and capillary).

$$BD = \frac{W_{ds}}{V}, \tag{1}$$

$$NCP = \frac{SWHC\ (\%) - CWC(\%)}{BD}, \tag{2}$$

$$CP = CMC\ (\%) \times BD, \tag{3}$$

$$TP = CP + NCP, \tag{4}$$

where V represents the ring knife volume; TP, CP, NCP, and BD are soil total porosity (%), soil capillary porosity (%), soil non-capillary porosity (%), and soil bulk density (g/cm³), respectively.

The soil water storage was measured from January to December 2024. Undisturbed soil samples were collected monthly from three forest types. First, a known volume of soil sample was weighed (W₁, g). The soil samples were oven-dried at 105 ℃ for 24 hours until constant weight. After drying, the soil sample was reweighed (W₂, g). The aluminum container was cleaned and weighed after removing the dried soil (W₃, g). The soil water storage was calculated as follows:





$$W = \frac{w_1 - w_2}{w_3} \times 100\% , \tag{5}$$

$$SWS = W \times BD \times h \times 10 , \tag{6}$$

Where W, SWS, and h are soil water content (%), soil water storage (mm), and soil thickness (cm); 10 is the conversion factor used to express the water layer in millimeters (mm).

**2.4 Statistical analysis**

Initial analysis using one-way ANOVA revealed significant variations in soil and hydrological properties among forest types at the same soil depth. When significant differences were detected ($P < 0.05$), the least-significant difference (LSD) test was applied for pairwise comparisons. Subsequently, two-way ANOVA was used to analyze the individual and interactive effects of forest type and soil depth on soil and hydrological characteristics. Furthermore, correlation analysis and redundancy analysis (RDA) were carried out to investigate relationships between soil and hydrological properties. The relative influence of soil parameters on hydrological characteristics were assessed using the "hier.part" and "vegan" packages. Data are expressed as mean ± SE, and all statistical computations were performed using R (version 4.2.1).

**3 Result**

**2.4 Soil particle size distribution across three forest types**

Soil texture analysis revealed sand content as the predominant particle fraction (mean 42.98%) throughout the 0-60 cm layer in all forest types, followed by silt (34.13%) and clay (22.89%) (Fig. 2). The soil particle size was significantly affected by the forest types. Specifically, the 0-60 cm clay content was higher in PF than in SF and RM, with significantly higher than in RM (P<0.05, Fig. 2a). The 0-60 cm silt content was significantly higher in RM than in PF and SF (P<0.05, Fig. 2c). The sand content in the 0-10 and 20-40 cm was significantly higher in PF than in RM (P<0.05, Fig. 2e). Overall, PF had the highest clay content, significantly higher than SF and RM (Fig. 2b). Meanwhile, sand content was significantly higher in both PF and SF than in RM (Fig. 2f).



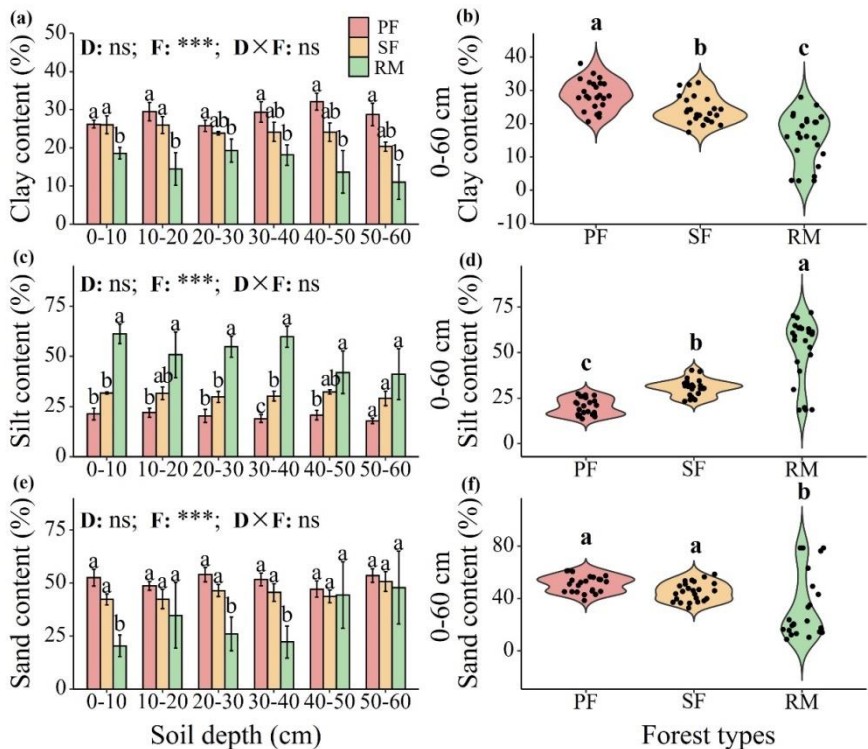

Figure 2 Soil particle size composition across three forest types. Different lowercase letters indicate statistically significant variations ($P < 0.05$) in soil properties at the same depth across forest types. PF: Primary Forests; SF: Secondary Forests; RM: Rubber Monocultures. D represents soil depth, F denotes forest types, and D×F signifies their interaction. ***: $P < 0.001$, ns: $P > 0.05$.

## 3.2 Soil properties across three forest types

Significant variations in soil properties were observed across the three forest types. The 0-20 cm BD was significantly greater in RM than in PF and SF. Conversely, SF exhibited higher BD than PF and RM in the 20-60 cm ($P<0.05$, Fig. 3a). The 0-60 cm TP was lower in SF than in PF and RM, with significant differences particularly in the 0-50 cm soil layer ($P<0.05$, Fig. 3b). The 0-60 cm CP in PF was significantly higher than in SF and RM ($P<0.05$, Fig. 3c). In contrast, the 0-60 cm NCP in RM was significantly higher than in PF and SF ($P<0.05$, Fig. 3d). Overall, most soil physical properties were significantly influenced by forest types and soil depth, except for depth effects on CP and NCP.

For soil chemical properties, the 0-60 cm SOM, TC, and TN were higher in PF than in SF and RM, especially significantly higher in the 0-20 cm ($P<0.05$, Fig. 4a-c), while C:N ratio was opposite (Fig. 4d). similarly, the 0-60 cm AN was significantly higher in PF than in SF and RM, and the AP showed for the same pattern in the 30-60 cm ($P<0.05$, Fig. 4e-f). Overall, most chemical properties were significantly influenced by forest types and soil depth, except for depth effects on AP



and C:N ratio.

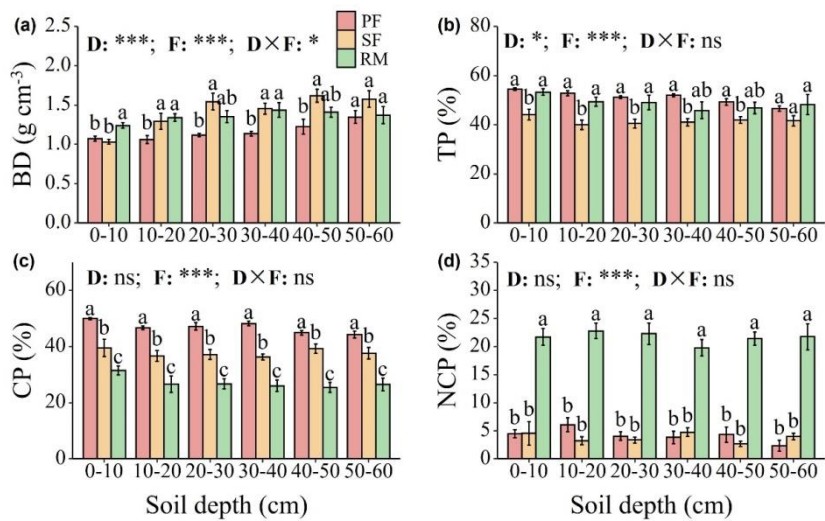

Figure 3 Soil physical properties across three forest types. BD (a), TP (b), CP (c), and NCP (d) are soil bulk density, soil total porosity, soil capillary porosity, and non-capillary porosity, respectively. D represents soil depth, F denotes forest types, and D×F signifies their interaction. ***: $P < 0.001$, *: $P < 0.05$, ns: $P > 0.05$.

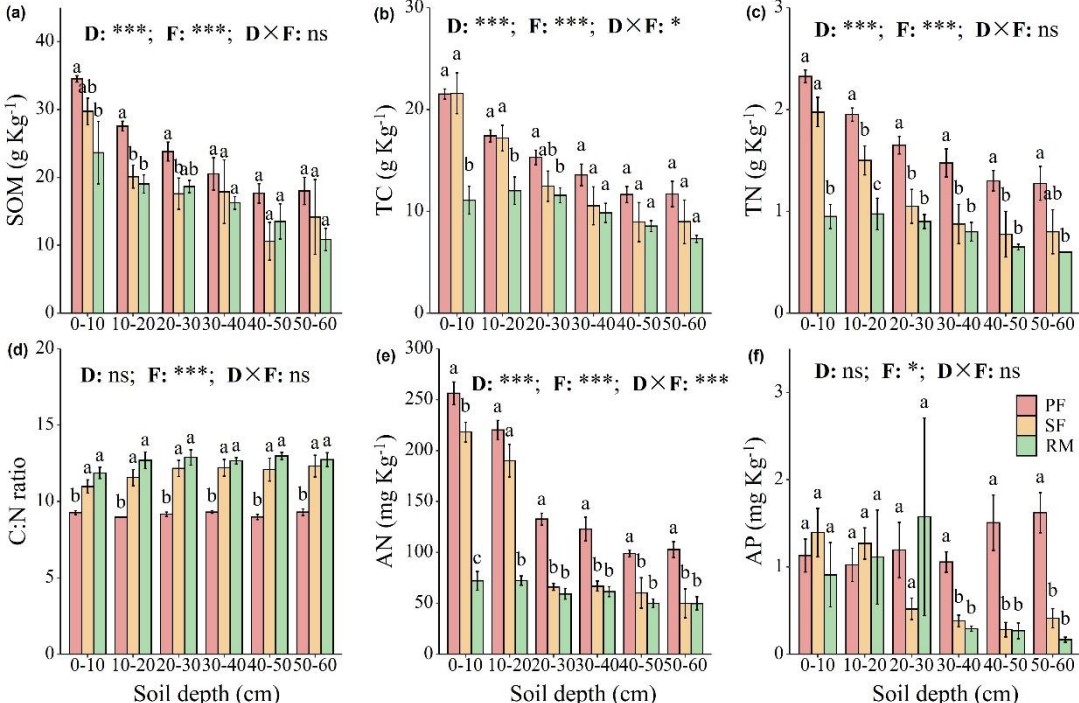

Figure 4 Soil chemical properties across three forest types. SOM (a), TC (b), TN (c), C:N ratio (d), AN (e), and AP (f) are soil organic matter, soil total carbon, soil total nitrogen, soil C: N ratio,



soil available nitrogen, and soil available phosphorus, respectively. D represents soil depth, F denotes forest types, and D×F signifies their interaction. ***: $P < 0.001$, *: $P < 0.05$, ns: $P > 0.05$.

### 3.3 Soil water retention across three forest types

Soil water retention was significantly affected by forest type ($P<0.05$, Fig. 5). Specifically, the 0-60 cm soil water retention in PF was significantly higher than in SF and RM ($P<0.05$, Fig. 5). In the 0-20 cm, SF showed significantly higher water retention than RM, with particularly significantly differences in the topsoil (0-10 cm). Conversely, RM had higher water retention than SF in the 20-40 cm, except for the SWC. Furthermore, Soil water storage showed significant seasonal variations across forest types, with higher values in the rainy season (May-October) than in the dry season (November-April) (Fig. 6). Overall, PF had significantly higher soil water storage than SF and RM. Additionally, the saturated hydraulic conductivity was higher in PF than in SF and RM (Table 2).

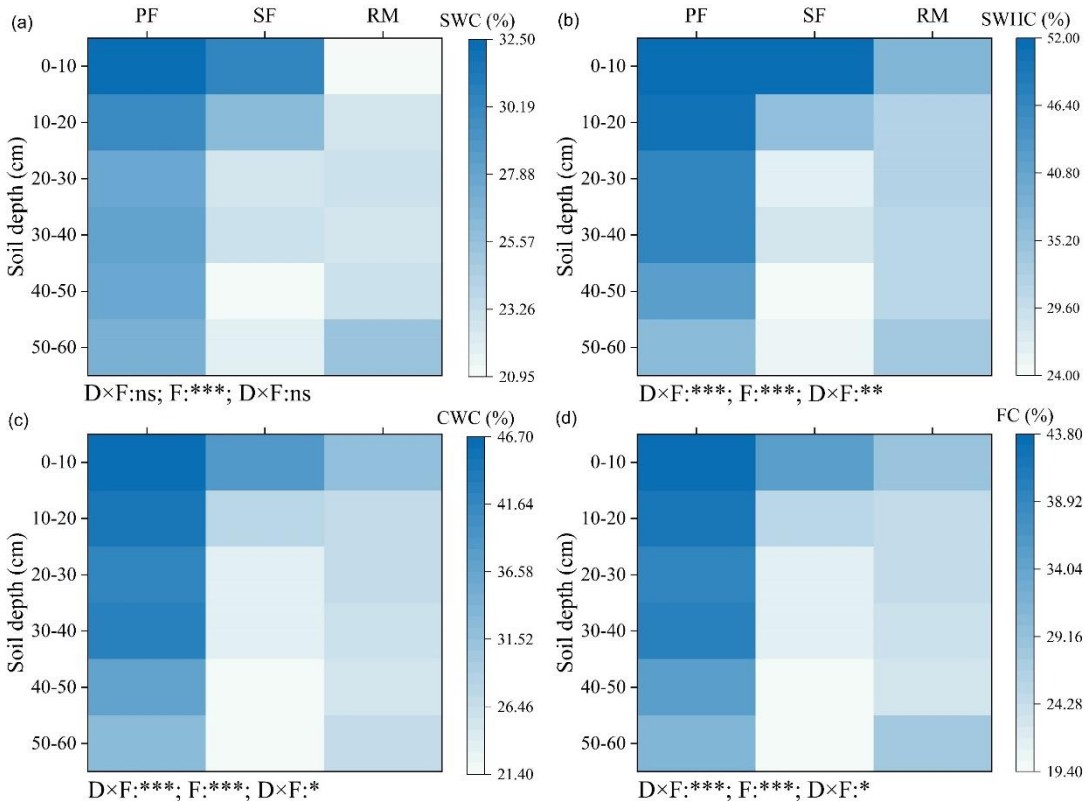

Figure 5 Soil water retention across three forest types. Different lowercase letters indicate statistically significant variations ($P < 0.05$) in soil properties at the same depth across forest types. The abbreviations SWC (a), SWHC (b), CWC (c), and FC (d) represents soil water content, saturated water holding capacity, capillary water capacity, and field capacity, respectively. ***: $P < 0.001$, **: $P < 0.01$, *: $P < 0.05$, ns: $P > 0.05$.





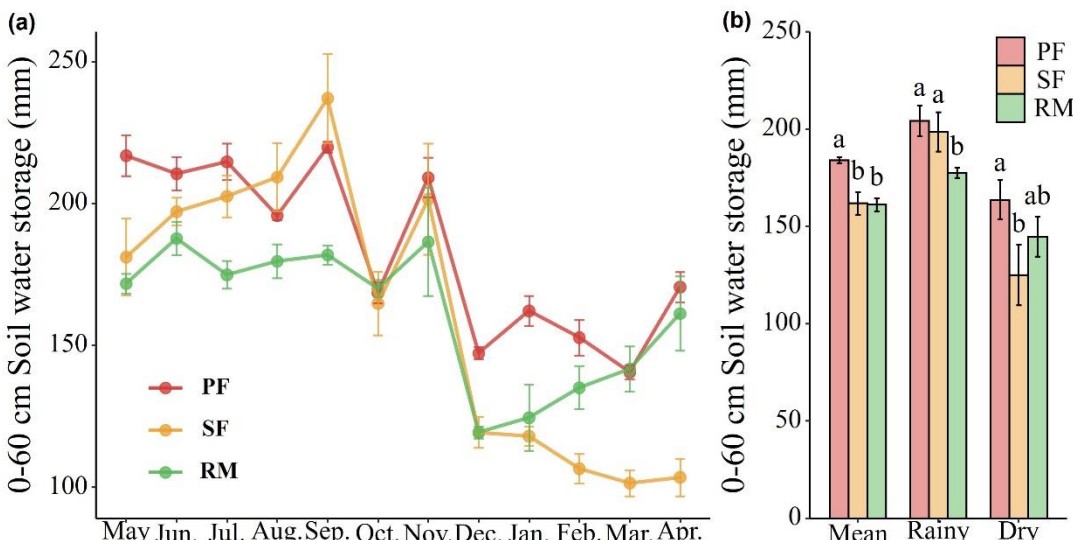

Figure 6 0-60 cm soil water storage across three forest types. (a): Monthly soil water storage of three forest types (Jan.-Dec. 2024; Rainy season: May-Oct., Dry season: Nov.-Apr.); (b): Monthly average soil water storage of the three forest types.

Table 2 Topsoil saturated hydraulic conductivity across three forest types

| Forest types | Primary forest | Secondary forest | Rubber monoculture |
|---|---|---|---|
| Saturated hydraulic conductivity (cm min$^{-1}$) | 71 | 27 | 2.43 |

## 3.4 Controlling factors affecting soil water retention

Both correlation analysis and RDA revealed significantly positive relationships between TP, CP, SOM, clay and sand content, and negative correlations with BD, NCP, and silt (Fig. 7-8). Soil water retention variability was 85.173% explained by RDA1 and RDA2 in combination (Fig. 8a). Among all soil properties, TP and CP collectively explained 62.10% of water retention variation (31.49% and 30.61%, respectively), followed by BD (12.46%) and NCP (7.69%). However, SOM and soil texture characteristics had relatively weak effects on water retention (Fig. 8b).



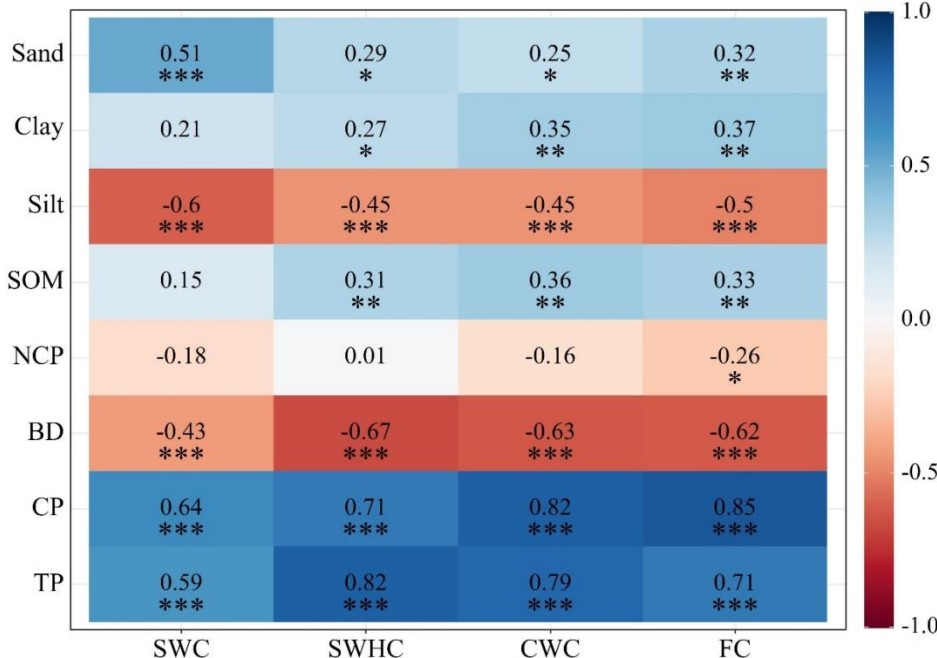

Figure 7 Spearman correlation heatmap between soil hydraulic properties and physicochemical characteristics across forest types. Color intensity indicates correlation strength. ***: $P < 0.001$, **: $P < 0.01$,*: $P < 0.05$.

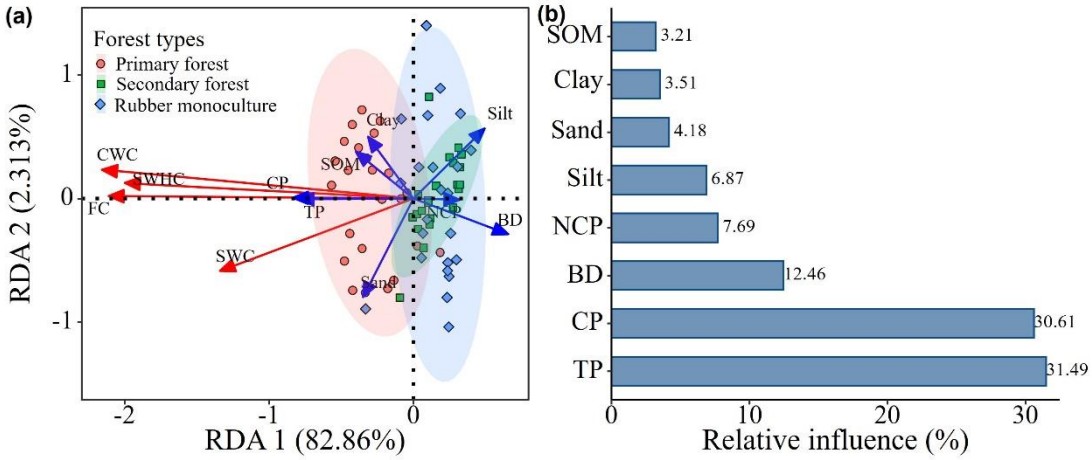

Figure 8 Relationships between soil properties and water retention capacity across forest types. (a) Redundancy analysis (RDA) and (b) relative influence of soil properties on soil water retention. Axis labels indicate percentage variance explained.





## 4 Discussion

### 4.1 Effects of forest conversion on soil properties

A great number of studies have found that the destruction of tropical primary forests causes a decline in soil quality (Sun et
al., 2021; Wen et al., 2019). For example, the conversion of primary forests to naturally restored secondary forests and
rubber monoculture resulted in an increase in soil bulk density (Jiang et al., 2023), and a decrease in soil fertility and soil
microbial richness (Lan et al., 2017). In our study, soil properties were found to be affected by forest conversion. Primary
forests had significantly better soil quality than secondary forests and rubber plantations with higher soil porosity and lower
soil bulk density. These results are in line with previous findings that the conversion of tropical primary forests to secondary
forests or rubber plantations resulted in a significant decrease in maximum water-holding capacity, total soil porosity, total
nitrogen, and soil organic carbon, but a significant increase in soil bulk density (Wen et al., 2019). Similarly, the topsoil bulk
density was lower in secondary forests compared to rubber monoculture, thus, secondary forests had higher total soil
porosity and capillary porosity (Fig. 3), which is consistent with Li et al. (2012), who found that soil bulk density was
significantly lower in tropical rainforests than in rubber plantations. The higher soil bulk density of rubber monoculture may
be related to anthropogenic activities, where frequent agricultural production activities compact the soil and increase the soil
bulk density (Guillaume et al., 2015, 2016). In addition, the lower soil clay content and higher silt content of rubber
monoculture exacerbated this result (Fig. 2). On the other hand, compared to rubber monoculture, primary and secondary
forests have higher litter content and complex root systems, especially in shallow soils. The inputs of litter decomposers and
root distribution increased soil porosity in primary and secondary forests, resulting in a decrease in soil compactness (Zhu et
al., 2021).

For chemical properties, we found that soil organic matter content was significantly higher in primary forests than in
secondary forests and rubber monoculture. In shallow soil, soil total carbon and total nitrogen were significantly higher in
primary forest than in rubber monoculture, but the difference between primary and secondary forests did not reach a
significant level (Fig. 4), which is consistent with the findings of Wen et al. (2019). The possible reason for this is that
primary forests have superior understory vegetation conditions and higher quality of litter organic matter inputs, thus
allowing more organic matter to be input into the soil through surface litter. In addition, fine root biomass is larger, turnover
is faster, soil microbial metabolism is relatively vigorous, and the amount of organic matter returned to the soil is higher
(Lorenz, 2021). Furthermore, the decomposition of litter is an important source of soil nutrient inputs. Compared to
secondary forests, rubber monocultures have a simpler root system and the surface litter is often cleared by humans to
230 facilitate rubber cutting, resulting in less litter on the surface, and, thus, less organic matter input from decomposition.

### 4.2 Effects of forest conversion on soil hydrological processes

Forest conversion affects soil hydrologic functions such as water-holding capacity, saturated water content, and capillary
water capacity by altering soil properties and vegetation structure (Mahe et al., 2005; Wang et al., 2021). We found that the





soil water retention of primary forests was significantly higher than that of naturally restored secondary forests and rubber

monoculture, which is consistent with previous study that soil water retention in secondary forests and rubber plantations
decreased by 27.7% and 11.5%, respectively, compared to natural forests (Wen et al., 2017). The soil water retention of
primary forests was significantly higher than that of the other two forest types, which was closely related to its high litter
yield and rich root structure. In our study, the litter yield of primary forests (4.27 t/hm$^2$) was significantly higher than that of
secondary forests (2.08 t/hm$^2$) and rubber monoculture (1.66 t/hm$^2$) (Table 1), which provided more organic matter input to

the soil, thus enhancing the soil water retention. In addition, we found that among soil properties, soil porosity and soil bulk
density were the key factors affecting soil water retention (Fig. 8). The lower soil bulk density and higher root density of
primary forests contribute to improved soil porosity, facilitating water infiltration and storage (Wen et al., 2017). However,
the topsoil water-holding capacity of secondary forests was significantly higher compared to that of rubber monoculture (Fig.
5). This was due to the high capillary porosity of the shallow soil in secondary forests, which facilitates rapid water

infiltration and storage, whereas the high bulk density of the deep soil restricts water infiltration and storage in the deep soil.
The water-holding capacity of the deeper soils of rubber monoculture was higher than that of secondary forests, due to the
higher total soil porosity and lower bulk density of the deeper soils in rubber monoculture. In addition, 70% of the water-
absorbing roots of rubber trees are located in the upper 30 cm of soil (Srinivasan et al., 2004), thus increasing the water-
holding capacity of deeper soils.

Soil water storage capacity serves as a critical indicator linking surface and subsurface water, reflecting the soil's ability to
absorb and store water. Primary forests, with their complex vegetation structure, rich litter layer, and high soil organic matter
content, are able to effectively intercept rainfall and reduce surface runoff, thereby increasing soil water infiltration and
storage (Puhlmann et al., 2007). In addition, the high soil organic matter content of primary forests also contributes to their
soil water retention, allowing them to maintain high soil water storage capacity during both the rainy and dry seasons. We

found no significant difference between secondary forests and rubber monoculture in monthly average water storage, but
their seasonal dynamics showed different characteristics. In the rainy season, the water storage capacity of secondary forests
was higher than that of rubber monoculture, which might be related to the vegetation structure and soil physicochemical
properties of secondary forests, as they could effectively intercept rainfall and increase the infiltration of soil water. However,
in the dry season, the soil water storage capacity of secondary forests was lower than that of rubber monoculture, which was

related to the shallower root distribution of secondary forests. The shallower root system of secondary forests prevents them
from utilizing deep soil water, leading to rapid water evaporation. In addition, rubber monoculture is more water-intensive
compared to native species, and the root system extension can explore a large portion of the soil to absorb water (Isarangkool
Na Ayutthaya et al., 2011). Depending on the season and vertical distribution of soil water, rubber trees can shift their water
uptake from shallow to deeper soils to extract water from the areas where it is most abundant at the time (Liu et al., 2014;

Maeght et al., 2015). Liu et al. (2014) found that rubber trees employ vertical root specialization to facilitate seasonal water
acquisition strategies. During the late rainy season when soil water is abundant, the functional roots primarily extracted
water from the upper 30 cm soil layer. As the dry season progresses, soil water content in the intermediate layer (30-70 cm)

become depleted, the water uptake focus gradually shifts to depths below 70 cm (Liu et al., 2014). This stratified utilization pattern effectively mitigates seasonal drought stress.

## 5 Conclusion

The study demonstrates that forest conversion in tropical primary forests significantly alters soil hydrological functioning. We found that primary forests exhibited significantly greater soil water retention compared to both secondary forests and rubber monocultures. Notably, secondary forests demonstrated superior water retention in surface soils (0-20 cm) relative to rubber plantations, whereas this relationship reversed in deeper soils. Primary forests displayed greater topsoil saturated hydraulic conductivity and water storage capacity than both secondary forests and rubber monocultures. Moreover, the water storage capacity in secondary forests and rubber monocultures displayed significant seasonal variations. We also found that soil physicochemical properties were affected by changes in forest type, and the conversion of primary forest to secondary forest and rubber monoculture increased soil bulk density and decreased soil nutrients. Soil water retention was primary affected by soil porosity. Our results suggest that the conversion of primary forests to secondary forests and rubber monoculture does not favor soil hydrological properties, which provides an important guideline for tropical rainforest management.

## Acknowledgments

The study was financially supported by National Natural Science Foundation of China (Grant No. 31922053), Key R&D Program of Hainan (Grant NO. ZDYF2022SHFZ042), the Second Tibetan Plateau Scientific Expedition and Research Program (Grant NO. 2019QZKK0405), and the start-up fund of Hainan University (Grant NO. KYQD(ZR)-22085, KYQD(ZR)21096).

## Author contributions

Zhongmin Hu and Licong Dai planned and designed the research. Qiaoyan Chen, Siyuan Cheng, Shuting Yu, Xiaowei Guo and Zhongyi Sun performed the experiments and analyzed the data. The first draft of the manuscript was written by Qiaoyan Chen and all authors commented on previous versions of the manuscript. All authors read and approved the final manuscript.

## Data availability

The raw data are available from the authors upon request.



**Conflict of Interest**

The authors declare that they have no conflict of interest.

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
