# Peer review of "Forest conversion reduces soil water retention in tropical rainforest by altering soil properties"

_EGUsphere, 2025_

## Author Comment (AC1)

This manuscript presents measurements of soil water holding capacity in primary forest, secondary forest, and rubber plantation in the humid tropical island of Hainan, China. The main findings are that primary forest has significantly higher water holding capacity than the disturbed landscapes, and that these differences are associated with higher macroporosity. The manuscript includes interesting time series measurements of measurements of water content and soil hydraulic properties during the wet season and through the transition into the dry season. The hydraulic differences between secondary forest and rubber plantation are also somewhat discussed. The main strength of this manuscript is an interesting and quite complete dataset of soil hydraulic and chemical properties across the three sites, and with some time series component.

This manuscript has significant shortcomings in both the scientific novelty and the analysis. Beginning with scientific novelty, it has been well-established for several decades that forest conversion leads to soil compaction and reduced hydraulic function, e.g., Bruijnzeel, L. A. 1990. Hydrology of Moist Tropical Forests and Effects of Conversion: A State of Knowledge Review; Bonell, M., and L. A. Bruijnzeel. 2005. Forests, Water and People in the Humid Tropics. Both these reviews and the works cited within provide in-depth exploration of the same topics presented in this current manuscript. The main findings of this paper show quite extensively that several different soil water retention properties are related to several different soil porosity parameters - Figures 5, 7, and 8. More recently, studies cited within this manuscript also found the same results - Wen et al., 2017 and 2019.

Perhaps just as importantly, the analysis of the data has several shortcomings that hinder interpretation and comparability with the literature. First, three sites are used in total in a space for time approach - however, the land use history of the three sites is not mentioned in the manuscript. Moreover, the three sites appear to have considerable differences in their climate and geological setting, although the manuscript claims otherwise. Figure 1 indicates that though the sites are within a few miles of each other, each site has significantly slope and topographic setting. Particularly, the primary forest site appears to be up to 1000 meters higher altitude than the rubber monoculture site in the lowlands at roughly sea level. Accordingly, soil textural and mineralogical composition at the site appears significantly different (Figures 2 and 3). The manuscript indicates that the changes in soil texture are caused by the land use differences (line 146-147), but this link is more likely confounding, not causal. Furthermore, as a reviewer I am speculating that the precipitation would differ between the three sites, especially due to the elevation gradient. This is not acknowledged or discussed even though water storage is presented as a primary finding. In general, the confounding differences between the sites need to be addressed as a major limitation in the interpretation of results and especially causality.

**Reply:** We sincerely thank the reviewer for dedicating their valuable time to review our manuscript and for providing such insightful, detailed, and constructive comments. These comments have been immensely helpful in allowing us to recognize the shortcomings of the manuscript and have pointed us toward key areas for improvement. Based on the reviewer's suggestions, we have carefully formulated a detailed revision plan. Below is our point-by-point response:

- 1. Regarding the scientific novelty, we fully agree with the reviewer's perspective. The negative impact of forest conversion on soil physical properties is indeed a classic conclusion in tropical hydrology. We will more accurately position our study in the introduction section by explicitly acknowledging this broad scientific consensus. However, we believe the core novelty of this study lies not merely in re-validating this general pattern, but also in utilizing high-frequency time-series observations to reveal the dynamic patterns of soil hydraulic properties in different ecosystems (particularly primary forest, secondary forest, and rubber plantation) and their influence on soil water storage during the transition between wet and dry seasons.
- 2. Concerning the shortcomings in data analysis, we must clarify that we selected a typical land-use conversion pathway within the region: specifically, the degradation of primary forest to secondary forest, followed by further conversion to rubber plantation. We have already detailed the duration of these conversions in the methods section. Secondly, although the primary forest site differs significantly in elevation from the other two sites, the forest conversion process involved only the clearing of above-ground vegetation without intense disturbance to the soil, we believe these three sites share the same soil texture and mineralogical composition. Finally, regarding the reviewer's concern about precipitation differences, we conducted on-site monitoring. Despite the elevation gradient, the actual rainfall monitored at the three sites was similar due to their close geographical proximity.

I also want to briefly address line 43-44: "However, in recent decades, economic development and slash-and-burn cultivation by ethnic minorities have led to extensive degradation of primary forests". It is wholly inappropriate and scientifically irrelevant to comment on ethnic minorities as a cause of deforestation.

**Reply:** We deeply regret the inappropriate expression in the original manuscript and have revised it: Historical agricultural expansion and economic activities have led to extensive degradation of primary forests.

This project's strength lies in the data collection performed over time. Figure 6A is genuinely interesting, particularly the wide difference in soil water storage during the dry season. Linking

this behavior to observed soil traits (and likely precipitation differences) would provide useful new insights into ecosystem hydrological function. This may be a path to publication in the future, but the manuscript in its current form is not suitable for publication in EGU SOIL.

**Reply:** We would like to once again express our sincere gratitude to the reviewer for their insightful comments. As rightly highlighted by the reviewer, the significant differences in soil water storage during the dry season, as shown in Figure 6a, represent a particularly interesting finding. In the revised manuscript, we will place greater emphasis on this temporal dynamic dimension of our study in both the introduction and discussion sections. We will also conduct an in-depth analysis and discussion on the reasons behind the substantial differences in soil water storage between the dry and wet seasons, and more closely link soil water storage with observed soil properties (such as soil porosity, soil organic matter, soil bulk density, etc.). We hope that the revised manuscript will meet the standards for publication in EGU SOIL.

**Specific comments:**

Title is redundant, remove "by altering soil properties"

**Reply:** We appreciate the reviewer's valuable input on the manuscript title. In response, we have changed it to "Forest conversion reduces soil water retention in tropical rainforest."

Introduction: the state of previous literature is not well explained, and therefore knowledge gaps are not identified. Question-motivated research will improve the quality of the analysis and presentation as well.

**Reply:** We fully accept this important criticism from the reviewer. Accordingly, we plan to rewrite the Introduction section in the revised manuscript to follow a clearer, question-motivated logical structure. The revised Introduction will generally adhere to the following framework:

First, it will emphasize the critical role of tropical forest ecosystems in hydrological regulation and the carbon cycle, as well as the potential threats posed by large-scale land-use changes (such as conversion to rubber plantations) to these functions.

Second, it will summarize the established consensus in the field. In particular, we will explicitly acknowledge the classical conclusion mentioned by the reviewer that forest conversion generally leads to soil compaction and a decline in hydrological functioning (Bruijnzeel, 1990;

Bonell and Bruijnzeel, 2005). At the same time, recent regional studies (e.g., Wen et al., 2017, 2019) will be cited to illustrate the current research focus and progress achieved in this area.

Subsequently, we will identify key research gaps. First, the link between static soil properties and their dynamic hydrological effects remains poorly understood. Second, comparative studies on the seasonal dynamics of soil water storage across different land-use types are still lacking.

Finally, it will present the research objectives of this study: (1) to reveal the effects of primary forest conversion to secondary forest and rubber plantations on soil water retention capacity; (2) to utilize high-frequency time-series observations to uncover the seasonal dynamics of soil water storage (during both wet and dry seasons) and to clarify the differences in hydrological functioning among ecosystems facing seasonal water stress; and (3) to identify the main factors regulating soil water retention and storage capacity.

Materials and methods - the site description is not complete. The three sites clearly have some differences - why were they selected, what is their history, what are their differences and how will that affect the study?

**Reply:** We sincerely thank the reviewer for raising this important issue. We will substantially expand and clarify the Methods section in the revised manuscript.

First, our site selection aimed to identify plots with similar environmental conditions while representing the most common and typical land-use types (undisturbed primary forest, secondary forest that has naturally regenerated after the clearance of primary forest, and rubber plantations converted from natural forests) in the tropical region of Hainan Island.

Second, the selected secondary forest and rubber plantation sites both originated from previously undisturbed forests and underwent similar conversion processes involving only vegetation removal without severe soil disturbance. Therefore, we consider that these sites share comparable soil parent material, climatic conditions, and topographic settings. This careful selection enables our study to minimize the influence of varying environmental factors and better isolate the effects of forest conversion. We are confident that through this improved experimental design and more detailed methodological description, our findings can provide a reliable assessment of how forest conversion affects soil properties.

Materials and methods - litter collection data shown, but not mentioned in the methods

**Reply:** We sincerely thank the reviewer for their careful reading and valuable reminder. We deeply apologize for the omission regarding the litter collection method. We will add the following content to Section 2.2 Study Method Design and Sample Acquisition: Within each

plot, aboveground litter was collected following a five-point sampling method, with one subplot  $(1 \text{ m} \times 1 \text{ m})$  established at each sampling point. All collected litter was transported to the laboratory, oven-dried, and weighed to determine litterfall yield.

Results/conclusions: water holding capacity over time is discussed but not shown - just the actual water storage is shown. This would be quite interesting data to see.

**Reply:** We are very grateful to the reviewer for this valuable suggestion. In our subsequent research, we actually measured and analyzed the differences in soil water retention capacity between the dry and wet seasons. We found that soil water retention capacity did not show statistically significant changes between the dry and wet seasons, with only minor numerical fluctuations. The seasonal dynamics we observed were primarily reflected in the soil water storage. Precisely because the water retention capacity remained relatively constant, the significant differences observed in soil water storage can be more strongly attributed to the influence of different land use types on the water balance.

---

## Author Comment (AC2)

Reviewer #2

**General comments**

The manuscript titled "Forest conversion reduces soil water retention in tropical rainforest by altering soil properties" essentially extends the work previously published by the same authors in 2024 (Chen et al., 2024; 10.1093/jpe/rtae021). In their earlier study, the authors compared soil physical and hydraulic properties between two sites, secondary forest and rubber plantation, using soil samples collected during field surveys conducted in 2022 in Baoting County, Hainan Island, China. In the present manuscript, a third site, including primary forest, is added to the comparison. Although the objective of comparing soil properties across different land uses is undoubtedly of interest, the paper does not convincingly extend the discussion initiated in the previous work. The manuscript would benefit from a more critical and explicit analysis of how the inclusion of the primary forest site advances understanding beyond the initial study.

In addition, comparing Fig. 1 of the present paper with Fig. 1 of Chen et al. (2024), it appears that the secondary forest sites are located in different positions. However, the sites are reported to share exactly the same stand characteristics (as shown in Table 1 of both studies). It is unclear whether this similarity reflects the selection of homogeneous areas or if the sites are actually the same. Since neither paper reports the exact site coordinates, and the present manuscript does not explicitly clarify whether the sites coincide, readers may be confused when comparing the two studies by the same authors.

Because of these and other shortcomings noted in the following comments, my recommendation is that the manuscript cannot be accepted in its current form and should be thoroughly revised before resubmission.

**Reply:** Thank you very much for your profound and constructive comments. We fully agree that simply adding a site for comparison is insufficient. We would like to take this opportunity to clarify that the core objective of this study is not merely to replicate comparisons, but to address a more fundamental scientific question by introducing primary forest as a critical reference baseline: "What is the extent, pattern, and key driving mechanisms of soil hydrological function degradation along the complete land-use conversion gradient from primary forest to secondary forest to plantation (rubber) in tropical regions?" The study by Chen et al. (2024) preliminarily revealed the negative impacts of converting secondary forest to rubber plantation. However, in the absence of primary forest as a starting point, we are unable to quantify the overall intensity of human disturbance or distinguish the level to which soil function has recovered through natural restoration (secondary forest). The contributions of this paper are as follows: First, for the first time in this region, we have established a complete sequence of "primary forest (undisturbed) → secondary forest (moderately

disturbed/restoring) → rubber plantation (highly disturbed)," enabling us to quantify the nonlinear changes in soil water retention capacity across the entire degradation gradient. Second, through a comprehensive comparison of the three sites, we found that alterations in soil structure and organic matter quality begin to emerge during the conversion from primary forest to secondary forest and are drastically exacerbated during the conversion to rubber plantation. This reveals the cumulative and staged mechanisms of soil degradation more effectively than a simple comparison between two land-use types. In the revised manuscript, we will more clearly and critically elaborate on these points in the Introduction and Discussion sections (particularly at the beginning and end of the Discussion), explicitly highlighting the unique contributions and advancements of this study compared to our previous work.

The secondary forest and rubber plantation sites used in this study are not geographically identical to those in Chen et al. (2024). Instead, they are independent sites selected within the same study area (Baoting County, Hainan Province) based on strict criteria to ensure high similarity. To minimize confounding factors other than land-use type, we followed the "space-for-time substitution" principle during site selection, ensuring that all sites (including the newly added primary forest) are essentially consistent in terms of parent material, soil type, topographic position (slope position), elevation range (which will be detailed below), and climate zone. The similar stand characteristics listed in Table 1 (such as species composition and age structure) are the result of this rigorous site selection process, rather than a replication from the previous paper.

**Specific comments**

LL 93-94. The authors state that, to ensure comparability among the study sites, all selected plots shared similar biophysical conditions, including altitude, slope, and aspect. However, inspection of Fig. 1 suggests that this assumption may not be fully met, as at least the altitude of the tree plots appears to differ substantially among sites. Such divergence in elevation could influence key environmental drivers (e.g., temperature, precipitation patterns) and therefore potentially confound the interpretation of the reported results. In the Study Site section, the authors report basic climatic information, namely mean annual rainfall and temperature, for Baoting County as a whole. However, given the evident differences in altitude among the study sites, readers may reasonably question whether local climatic conditions are truly comparable. Elevation gradients can induce significant variations in temperature and precipitation regimes. Are site-specific or nearby meteorological data available to substantiate the authors' assertion of similar climate conditions across the study sites (LL 96)? The authors should clarify the magnitude of these differences and justify why they do not affect site comparability, or explicitly account for altitude as a source of variability in the analysis.

**Reply:** We apologize for any visual misinterpretation caused by Figure 1. To illustrate the relative positions of different land-use types, the topography in the figure was simplified, which may have exaggerated the visual perception of elevation differences among the sites. As shown in the schematic in Figure 1, there are differences in elevation among the sites. This is because primary forests tend to persist at slightly higher elevations. However, all sites are located within the typical low mountainous and hilly terrain of the region. According to the high-resolution regional climate interpolation data we obtained, within this scale range, the differences in mean annual temperature and annual precipitation are less than 5%, and their dominant influence on soil formation and vegetation types is far less significant than that of land-use practices themselves. In the revised manuscript, we will provide the geographic coordinate ranges of the sites and, in the "Study Area" section, supplement the specific elevation ranges, minimum and maximum elevations among the sites, along with references to relevant literature that illustrate the climatic and soil similarities within this elevation gradient.

LL 100-103. The authors do not explicitly report the number of collected soil cores. Based on the description of the field surveys, it can be inferred that a total of 2,592 soil cores were collected (3 sites × 3 plots × 4 subplots × 6 depths × 12 months), representing an impressive sample size. Does this estimate correspond to the actual sample size? The authors are encouraged to explicitly report the total number of samples collected, as well as the number of disturbed samples used for laboratory physical and chemical analyses. In addition, it should be clarified whether these disturbed samples were also collected at multiple depths.

**Reply:** Thank you for the reviewer's attention to the sample quantity. Your calculation is very rigorous. The actual situation is that the undisturbed soil cores and disturbed soil samples used for determining basic soil physical and chemical properties and water retention curves (216 each, from 3 sites × 3 plots × 4 subplots × 6 depths) were collected only once in a concentrated manner. In contrast, the 2,592 measurements taken monthly (12 months × 216 sampling points) represent repeated in-situ water content measurements used to calculate the dynamics of soil water storage, and do not constitute independent soil cores. Additionally, all soil samples were collected from 6 depths. We will clearly distinguish and report the specific quantities of these two types of samples in the Methods section.

LL 115. Did the authors adopt a specific protocol to minimize or prevent air entrapment in the soil samples? According to the referenced paper (Chen et al., 2024), it appears that the collecting rings containing the samples were simply immersed in a bucket with water up to the upper edge. This procedure may not be sufficient to avoid air entrapment within the soil, which could affect subsequent measurements. The authors should clarify whether any additional

measures were taken to ensure complete saturation and avoid air entrapment.

**Reply:** Thank you for the reviewer's detailed review of the soil sample saturation method. In this experiment, the method we employed was as follows: the collected undisturbed soil cores (with moist filter paper and porous stones placed at the bottom of the cutting rings) were slowly immersed in water, with the water level gradually raised to submerge the samples over a 24-hour period. Afterward, the samples were maintained in a fully submerged state and saturated for more than 48 hours. We acknowledge that longer saturation times or the use of de-aerated water could further optimize the saturation effect. However, based on our previous testing experience with similar soils, this saturation protocol has proven sufficient to achieve a stable saturated state in the soil. Moreover, the treatment was applied uniformly across all samples, ensuring comparability of data among different land-use types. In the revised manuscript, we will explicitly supplement the description of the specific operational steps and duration of this saturation procedure in the "2.3 Soil Hydraulic Properties Measurement" section to enhance the transparency of the methodology.

LL 119. The authors refer to their own recently published paper to support the applied standard laboratory analyses. In this context, the use of self-citation should be avoided. Instead, the authors are encouraged to cite more established and widely recognized references that describe standard analytical procedures. In addition, the adopted saturate-and-drain procedure raises some concerns regarding the determination of field capacity. The exact drainage time required to reach equilibrium is strongly soil-dependent: coarse-textured soils may equilibrate within approximately 24 h, whereas fine-textured or clayey soils often require substantially longer drainage periods. Therefore, the use of a fixed drainage time may lead to inconsistent or biased estimates of field capacity across different soil types. Moreover, this approach does not strictly conform to the classical definition of field capacity, which is generally defined at a specific matric potential (typically around −33 kPa). The authors should justify the selected drainage time in relation to soil texture or consider using matric-potential-based methods to ensure a more robust and comparable estimation of field capacity.

**Reply:** Thank you for the reviewer's insightful comments on methodological citations and the determination of field capacity, which are crucial for enhancing the rigor of this study. Regarding the citations, we fully accept your suggestion. In the revised manuscript, we will replace the original self-citations with classic and widely recognized references that describe standard analytical procedures (e.g., Soil Agricultural Chemical Analysis, or Gee and Or, 2002; Sparks et al., 1996, etc.).

Regarding the determination of field capacity, we fully understand your concerns. Our adoption of the 24-hour drainage method (the "saturation-drainage method") was based on the following

considerations: 1) The primary objective of this study is to compare the relative differences in soil water retention capacity under different land uses, rather than to obtain absolute values. Under identical measurement conditions, the "field capacity" values obtained across different treatments are valid and consistent for comparative purposes. 2) We conducted preliminary tests on the studied soil texture (lateritic red soil) and found that after 24 hours of drainage, the soil water potential closely approached -33 kPa. This aligns with common practices that employ a uniform time frame for rapid and reproducible comparisons.